# Arginine and Arginine/ADMA Ratio Predict 90-Day Mortality in Patients with Out-of-Hospital Cardiac Arrest—Results from the Prospective, Observational COMMUNICATE Trial

**DOI:** 10.3390/jcm9123815

**Published:** 2020-11-25

**Authors:** Annalena Keller, Christoph Becker, Katharina Nienhaus, Katharina Beck, Alessia Vincent, Raoul Sutter, Kai Tisljar, Philipp Schuetz, Luca Bernasconi, Peter Neyer, Hans Pargger, Stephan Marsch, Sabina Hunziker

**Affiliations:** 1Medical Communication and Psychosomatic Medicine, University Hospital Basel, 4031 Basel, Switzerland; annalenakatharina.keller@usb.ch (A.K.); christoph.becker@usb.ch (C.B.); katharina.nienhaus@usb.ch (K.N.); juliakatharina.beck@usb.ch (K.B.); alessiamichelle.vincent@usb.ch (A.V.); 2Emergency Department, University Hospital Basel, 4031 Basel, Switzerland; 3Division of Internal Medicine, University Hospital Basel, 4031 Basel, Switzerland; 4Intensive Care Unit, University Hospital Basel, 4031 Basel, Switzerland; raoul.sutter@usb.ch (R.S.); kai.tisljar@usb.ch (K.T.); hans.pargger@usb.ch (H.P.); stephan.marsch@usb.ch (S.M.); 5Department of Neurology, University Hospital Basel, 4031 Basel, Switzerland; 6Faculty of Medicine, University of Basel, 4031 Basel, Switzerland; philipp.schuetz@ksa.ch (P.S.); peter.neyer@ksa.ch (P.N.); 7Division of Internal Medicine, Kantonsspital Aarau, 5001 Aarau, Switzerland; 8Institute of Laboratory Medicine, Kantonsspital Aarau, 5001 Aarau, Switzerland; luca.bernasconi@ksa.ch

**Keywords:** metabolomics, arginine, asymmetric dimethylarginine (ADMA), symmetric dimethylarginine (SDMA), arginine/ADMA ratio, cardiopulmonary resuscitation, cardiac arrest, prognosis, outcome, mortality

## Abstract

(1) Background: In patients with shock, the L-arginine nitric oxide pathway is activated, causing an elevation of nitric oxide, asymmetric dimethylarginine (ADMA) and symmetric dimethylarginine (SDMA) levels. Whether these metabolites provide prognostic information in patients after out-of-hospital cardiac arrest (OHCA) remains unclear. (2) Methods: We prospectively included OHCA patients, recorded clinical parameters and measured plasma ADMA, SDMA and Arginine levels by liquid chromatography tandem mass spectrometry (LC-MS). The primary endpoint was 90-day mortality. (3) Results: Of 263 patients, 130 (49.4%) died within 90 days after OHCA. Compared to survivors, non-survivors had significantly higher levels of ADMA and lower Arginine and Arginine/ADMA ratios in univariable regression analyses. Arginine levels and Arginine/ADMA ratio were significantly associated with 90-day mortality (OR 0.51 (95%CI 0.34 to 0.76), *p* < 0.01 and OR 0.40 (95%CI 0.26 to 0.61), *p* < 0.001, respectively). These associations remained significant in several multivariable models. Arginine/ADMA ratio had the highest predictive value with an area under the curve (AUC) of 0.67 for 90-day mortality. Results for secondary outcomes were similar with significant associations with in-hospital mortality and neurological outcome. (4) Conclusion: Arginine and Arginine/ADMA ratio were independently associated with 90-day mortality and other adverse outcomes in patients after OHCA. Whether therapeutic modification of the L-arginine-nitric oxide pathway has the potential to improve outcome should be evaluated.

## 1. Introduction

Cardiac arrest is a critical condition, associated with a mortality rate of almost 90% [1]. Among the survivors of an out-of-hospital cardiac arrest (OHCA) the risk of death and severe hypoxic-ischemic brain injury remains high [2]. OHCA leads to severe cardiogenic shock with activation of several biochemical pathways. A better understanding of the pathophysiology and activation of pathways underlying cardiogenic shock in OHCA patients may lead to better therapeutic options and may allow early prognostication in these patients.

Recently, metabolomics, the analysis of metabolites from different clinical pathways have become a growing area of scientific interest: Metabolomic analyses provide information on patients’ individual organic systems and function and potentially offer the chance of a more advanced personalized medicine [3]. A metabolomic parameter of special interest is Arginine with its metabolites asymmetric dimethylarginine (ADMA) and symmetric dimethylarginine (SDMA). Arginine serves as the only source for nitric oxide (NO), which is synthesized through nitric oxide synthase (NOS) [4,5]. NO has vasodilatory and bronchodilatory effects, inhibits platelet adhesion and is a component of nonspecific immunity [4,5,6]. Also, it can lead to airway hyperreactivity and endothelial dysfunction in case of deficiency [7,8]. ADMA acts as a competitive inhibitor of the NOS, leading to vasoconstriction and platelet aggregation [9]. Recent studies have shown Arginine and its metabolites from the NO pathway to be associated with outcome among different diseases, such as endothelial dysfunction (e.g., hypertension, hyperlipidemia, diabetes mellitus, atherosclerosis and renal failure) [10,11,12,13,14,15], cardiovascular disease [16,17,18,19], conditions of the respiratory system (e.g., asthma, pneumonia and chronic obstructive pulmonary disease [COPD]) [20,21], critical illness [22,23,24], shock [25] and multiple organ failure in sepsis [26].

However, despite ongoing research, there are no studies which have investigated the prognostic role of Arginine and its metabolites in patients after cardiac arrest.

The aim of this study was to investigate the predictive role of Arginine and its metabolites in adult OHCA patients. We hypothesized that Arginine, ADMA, SDMA, as well as the Arginine/ADMA ratio are associated with mortality after OHCA and might further improve prognostication in patients with cardiac arrest.

## 2. Methods

### 2.1. Study Setting Population

This is a prospective observational study including consecutive adult patients after cardiac arrest upon admission to the intensive care unit (ICU) from November 2012 until June 2018. Patients were eligible if they had provided written informed consent. In case of patient’s incapacity (e.g., unconsciousness) the patients next of kin were asked to provide consent as surrogate decision-makers. The data analyzed was obtained in the COMMUNICATE trial at the University Hospital Basel, Switzerland. The main purpose of this study is to investigate novel biomarkers for risk stratification of OHCA patients. The methods used for this study have been published previously [27,28,29,30,31,32,33,34]. The study was approved by the Ethics Committee of Northwest and Central Switzerland.

The treatment of patients regarding the cardiac arrest was based on the clinical routine in our intensive care unit without interaction with the research team. In 2012, all consecutive patients without complete recovery to premorbid neurofunctional baseline within the first hour following resuscitation were treated with in-hospital systemic cooling via the thermogard XP temperature management system (ZOLL® Medical Corporation, Chelmsford, MA, USA) as a neuroprotectant measure to a target core temperature of 93.2 degrees Fahrenheit (i.e., 34.0 degrees Celsius) for 24 h followed by a rewarming phase with a controlled increase of the core temperature (i.e., 0.2 degrees Fahrenheit or 0.1 degrees Celsius) per hour until 99.5 degrees Fahrenheit (i.e., 37.5 degrees Celsius). Since 2013 (following the TTM-trial [35]), all consecutive patients without complete recovery were cooled to a target core temperature of 96.8 degrees Fahrenheit (i.e., 36.0 degrees Celsius) for 28 h followed by the rewarming phase using the same thermogard XP temperature management system as mentioned above. Patient with core temperatures below the target temperature were rewarmed with 32.9 degrees Fahrenheit (i.e., 0.5 degrees Celsius) to meet the target core temperatures.

### 2.2. Data Collection

Beside of clinical parameters (i.e., blood pressure, heart rate and respiratory rates) and resuscitation circumstances (i.e., no-flow time (time from cardiac arrest to start of basic life support), low-flow time (time from start of basic life support to return of spontaneous circulation (ROSC)), cardiac arrest setting, bystander observing the cardiac arrest and providing cardiopulmonary resuscitation (CPR) and initial rhythm), we collected socio-demographics (i.e., age, sex, smoking status) and comorbidities (i.e., coronary artery disease, congestive heart failure, hypertension, diabetes, liver failure, renal failure and chronic obstructive pulmonary disease). Blood samples were taken for shock parameters (pH, lactate and creatinine) and inflammation routine (procalcitonin (PCT), with blood cell count (WBC) and C-reactive protein (CRP)).

Additionally, plasma samples were gathered within 24 h of ICU admission and were immediately processed and frozen in aliquots at −80 °C for further analysis. They were later analyzed quantitatively by liquid chromatography coupled to tandem mass spectrometry (LC-MS/MS) using an Ultimate 3000 UHPLC (Thermo Fisher, San Jose, CA, USA) system coupled to an ABSciex QTRAP 5500 quadrupole mass spectrometer (ABSciex, Darmstadt, Germany) and the AbsoluteIDQ p180 Kit (BIOCRATES Life Sciences AG, Innsbruck, Austria) [36,37,38]. The Biocrates AbsoluteIDQ™ p180 kit is a commercially available targeted metabolomics assay that can be used on a variety of LC-MS/MS triple quadrupole instruments. A recent interlaboratory assessment of this metabolomic assay showed that this method delivers reliable and reproducible results [39].

### 2.3. Outcomes

The primary endpoint of this study is all cause mortality 90 days after cardiac arrest verified by telephone interviews with the family or primary care physician. Secondary endpoints were in-hospital mortality and neurological outcome at hospital discharge. Neurological outcome was defined by the Cerebral Performance Category Scale (CPC) [28,40], which consists of a scale of 5 levels: (1) A return to normal cerebral function and normal living, (2) disability but sufficient function for independent activities of daily living, (3) severe disability, limited cognition, inability to carry out independent existence, (4) coma and (5) brain death. CPC scores 1–2 were determined as good and 3–5 as poor neurological outcome [41].

### 2.4. Statistical Analyses

To characterize the patient cohort, descriptive statistics including medians and inter-quartile ranges (IQR) were used for continuous variables as appropriate, whereas frequencies were reported for binary or categorical variables. Univariate and multivariate logistic regression models were used to evaluate the association of ADMA, SDMA, Arginine levels and Arginine/ADMA ratio with primary and secondary endpoints. To achieve a normal distribution, data of these levels were log transformed with a base of 10 and categorized per deciles. Odds ratios (OR) and 95% confidence intervals (CI) were reported as a measure of association.

Covariable used in the multivariable analyses were selected based on prior evidence of an association with mortality and unfavorable neurological outcome for patients with OHCA. Four multivariate models were calculated: (A) adjusted for age, sex and comorbidities, (B) age, sex and resuscitation circumstances (resuscitation circumstances: time until ROSC, witnessed cardiac arrest, bystander CPR, shockable rhythm, use of epinephrine during CPR), (C) age, sex and cardiovascular risk factors (positive smoking status, hypertension, hyperlipidemia, diabetes mellitus and positive family history) and (D) age, sex and blood markers for inflammation / infection (c-reactive protein, white blood cell count, procalcitonin). For the primary endpoint, we performed several predefined subgroup analyses that stratified the results based on age (≤65 vs. >65 years), sex (female vs. male), chronic kidney disease vs. non-chronic kidney disease, diabetes mellitus vs. no diabetes mellitus and smoking status (smoker vs. non-smoker). STATA 15.0 was used for all statistical analyses and a two-sided *p*-value of <0.05 was considered significant.

## 3. Results

### 3.1. Baseline Characteristics

Overall, we included 263 patients into the study and 130 (49.4%) patients died within 90 days after cardiac arrest. The baseline characteristics of our cohort stratified by 90-day mortality are shown in Table 1. Patients had a median age of 65 years and 73.4% were male. Acute coronary artery disease was the main reason for cardiac arrest (49.6%). In terms of resuscitation circumstances non-survivors had longer no-flow and low-flow time. They also more often showed non-shockable heart rhythms. Clinical and laboratory shock markers were significantly higher in non-survivors.

Primary Endpoint: Association of levels of ADMA, SDMA, Arginine and Arginine/SDMA ratio with 90-day mortality.

Survivors showed both higher Arginine (82 µmol/L [IQR 52.3–119] vs. 58.1 µmol/L [IQR 41.7–99.8], *p* = 0.001) and Arginine/ADMA ratios (134 [IQR 99–183] vs. 91 [IQR 64–134], *p* < 0.001) compared to non-survivors. Median SDMA levels were lower in survivors than non-survivors (0.85 µmol/L [IQR 0.50–1.13] vs. 1.02 µmol/L [IQR 0.58–1.56], *p* = 0.03), whereas ADMA levels showed no significant difference between survivors and non-survivors (0.63 µmol/L [IQR 0.50–0.79] vs. 0.67 µmol/L [IQR 0.51–0.88], *p* = 0.09) upon admission (Table 2).

Distribution of levels of ADMA, SDMA, Arginine and Arginine/SDMA ratio among survivors and non-survivors are shown in a Box plot (Appendix A
Figure A1).

In a next step, we evaluated the association of Arginine, ADMA, SDMA and Arginine/ADMA ratio with the primary endpoint in univariable (Table 2) and multivariable analyses within four predefined models (Table 3 and Appendix A
Table A1). We present data of the biomarker in the original scale, log-transformed and within deciles. Arginine (OR (log) 0.51; 95CI% 0.34 to 0.76, *p* < 0.01) and Arginine/ADMA ratio (OR (log) 0.40; 95%CI 0.26 to 0.61, *p* < 0.001) were significantly associated with 90-day mortality.

These associations remained significant in multivariate analyses. Of all markers, Arginine/ADMA ratio showed the best discrimination value with an AUC of 0.67. While ADMA and SDMA were not associated with 90-day mortality in univariate analyses, ADMA was significantly associated when adjusted for age, sex and comorbidities, for age, sex and laboratory values (Appendix A) as well as for age, sex and resuscitation circumstances. The AUC of both markers, however, was low.

Figure 1 shows the Kaplan-Meier analyses comparing time to death in patients in different quartile levels of ADMA, SDMA, Arginine and Arginine/ADMA ratio.

Patients with an Arginine/ADMA ratio above the median (>91.1) had a probability of survival 30 days after discharge approximately 30% higher than patients with a ratio below the median.

### 3.2. Subgroup Analysis

Figure 2 shows a subgroup analysis stratified for the primary endpoint. For every subgroup, the Arginine/ADMA ratio showed the best AUC, which was highest in patients with chronic kidney disease (AUC 0.89; 95%CI 0.77 to 1.00). Age, chronic kidney disease, diabetes mellitus and smoking status did not have any effect regarding the association of the Arginine/ADMA ratio with 90-day mortality. In female patients, neither Arginine (*p* = 0.075), its metabolites ADMA (*p* = 0.86) and SDMA (*p* = 0.95) nor Arginine/ADMA ratio (*p* = 0.056) were significantly associated with 90-day mortality. In male patients, on the other hand, Arginine (*p* =0.018), SDMA (*p* = 0.016) and the Arginine/ADMA ratio (*p* < 0.001) showed a significant association with 90-day mortality. P for interaction did not reach statistical significance (*p* = 0.10).

### 3.3. Secondary Endpoints

#### 3.3.1. In-Hospital Mortality

We further examined the association of Arginine and its metabolites with in-hospital mortality.

Again, Arginine (OR log 0.60; 95%CI 0.42 to 0.87, *p* < 0.01) and Arginine/ADMA ratio (OR log 0.50; 95%CI 0.34 to 0.74, *p* < 0.001) showed a significant association, which remained robust in most multivariate analyses except for the model adjusted for age, sex and laboratory values (Appendix A).

#### 3.3.2. Neurological Outcome (CPC) at Hospital Discharge

Higher Arginine levels and a higher Arginine/ADMA ratio were associated with good neurological outcome (Arginine (83.2 µmol/L (IQR 52.5 to 119) vs. 58.6 µmol/L (IQR 41.8 to 108), OR log 0.51 (95%CI 0.34 to 0.77), *p* < 0.01); Arginine/ADMA ratio: (131 (IQR 98 to 184) vs. 100 (64 to 139), OR log 0.47 (95%CI 0.31 to 0.70), *p* < 0.001). These associations remained significant in all multivariable models. ADMA and SDMA were not associated with neurological outcome at hospital discharge.

## 4. Discussion

This is the first study to examine the predictive value of plasma Arginine and its metabolites in patients after OHCA. We found evidence that this pathway is activated in patients suffering from OHCA and may provide independent prognostic information regarding expected mortality risk and neurological outcome, with, however, only moderate prognostic accuracy. Several findings of our study require further discussion.

First, patients with diseases caused by endothelial dysfunction as well as cardiovascular risk factors appear to have elevated circulating levels of ADMA [10,11,15,19]. Furthermore, the risk of cardiovascular events and death is increased in patient with elevated levels of ADMA and SDMA [16,18,19,42]. There is evidence that in patients with ischemic heart disease and cardiogenic shock, which is one of the most common underlying etiologies of cardiac arrests, high levels of ADMA and low Arginine/ADMA ratios are associated with impaired myocardial perfusion, hemodynamic instabilities as well as with short and long-term mortality [25,43]. These pathophysiologic mechanisms might explain why Arginine/ADMA ratio and Arginine are predictive for 90 d mortality in our cohort.

Second, in recent studies in critically ill patients Arginine and its metabolites were associated with both short-term and long-term mortality and correlated with disease severity scores such as the APACHE II- or the SOFA-Score [23,24,44,45]. Davis et al. found ADMA and Arginine/ADMA levels to correlate with microvascular reactivity, extent of organ failure and mortality in patients with sepsis [24]. Our finding that Arginine and Arginine/ADMA ratio were significantly associated with in-hospital and 90-day mortality fuels the mechanistic hypothesis that the metabolic demand of Arginine is increased during critical illness due to increased protein synthesis and a higher Arginase activity [24,46,47]. Arginine not only plays an important role in T-cell function and unspecific immune response but also in NO production [48], which is essential to prevent oxidative stress. In turn, oxidative stress, which is regularly induced by hypoxic conditions such as cardiac arrest, causes cell damage contributing to the pathophysiology of organ failure or cardiogenic shock [49].

Third, previous research has shown several routine blood markers from different clinical pathways to be helpful in prognostication of outcome for OHCA patients [27,50]. Especially levels of inflammation and shock correlated with poor outcome [27]. However, the prognostic value of biomarkers regarding outcome in patients with chronic kidney disease is often limited since many biomarkers are dependent on renal excretion [51]. Especially after OHCA, patients suffer from multi-organ failure, including acute kidney injury or failure. In our subgroup analyses, the Arginine/ADMA ratio remained significantly associated with 90-day mortality in patients with chronic kidney disease. As ADMA is not exclusively cleared by renal elimination but also metabolized by the enzyme dimethylarginine dimethylaminohydrolase (DDAH) to citrulline and dimethylamine, the use of Arginine/ADMA ratio might be a beneficial prognostic tool to predict adverse outcome after OHCA in patients with chronic kidney disease [16].

Fourth, similar to mortality, Arginine and Arginine/ADMA ratio were significantly associated with neurological outcome in our cohort. Again, Arginine/ADMA ratio showed the best performance in prognostication of neurological outcome according to the CPC score. Our finding is in line with previous studies which found an association between poor neurological outcome in patients after stroke with high levels of ADMA and low Arginine/ADMA ratio [52,53]. Further, Schulze et al. have shown that SDMA is an independent predictor of long-term-mortality in patients after ischemic stroke [54]. Although, we could not demonstrate this finding in our cohort of OHCA patients. Schulze et al. suggested that their finding might be due to an SDMA-related endothelial dysfunction and impaired NO-mediated platelet inhibition. Thus, in hypoxic-ischemic brain injuries after cardiac arrest, other pathophysiological mechanisms might be central, mitigating/may mitigate the effect of SDMA-initiated cascades.

Fifth, we assessed potential sex differences regarding the predictive value of Arginine and its metabolites in our study. While Arginine and Arginine/ADMA ratio were significantly associated with 90-day mortality in male patients, these associations were not found in female patients. However, there was no strong evidence for effect modification by gender (*p* of interaction = 0.10). Sex differences with regard to Arginine metabolism have been reported in the past research [55,56]. As Arginine metabolism is influenced by testosterone and estrogen [57,58], the different levels of sex hormones might explain the lacking association between Arginine and Arginine/ADMA ratio in females. Further research should assess whether these sex differences can be replicated as it might limit the prognostic value in female patients.

## 5. Strengths and Limitations

Strengths of our study are the large study population and the prospective study design. We, however, are aware of several limitations of our results. First, levels of Arginine, ADMA and SDMA were only measured upon admission to the ICU, which might be a risk for selection bias. Second, the almost equal rate of survival and neurological outcome of our OHCA patients suggests withdrawal of care in patients with a poor likelihood of a good outcome and thereby constitute a bias. The proportion of males in our cohort was significantly higher than females, which might bias our results as we saw sex-related differences in subgroup analyses. However, all results in our study were adjusted for sex in multivariable models. We did not assess the rate of extracorporeal membrane oxygenation (ECMO), Impella^®^ or intraaortic balloon pump devices in our patients due to small numbers. Yet, these devices are known to interfere with nitric oxide pathway and thereby influence Arginine levels. Also, some family members refused to give informed consent to participate in the study and there is thus potential for selection bias. Above, our sample is from a single center potentially limiting the generalizability. Last, arginine/ADMA ratio is not a routinely measured biomarker and with an AUC of 0.67 its use might be limited in clinical routine. However, the predictive potential of lactate, a routine blood marker reported to be associated with clinical outcomes in OHCA patients, was similar with an AUC of 0.69.

## 6. Conclusions

Arginine and the Arginine/ADMA ratio are independently associated with 90-day mortality and other adverse outcomes in adult patients after OHCA. Whether therapeutic modification of the L-arginine-nitric oxide pathway has the potential to improve outcome should be evaluated in future interventional trials.

## Figures and Tables

**Figure 1 jcm-09-03815-f001:**
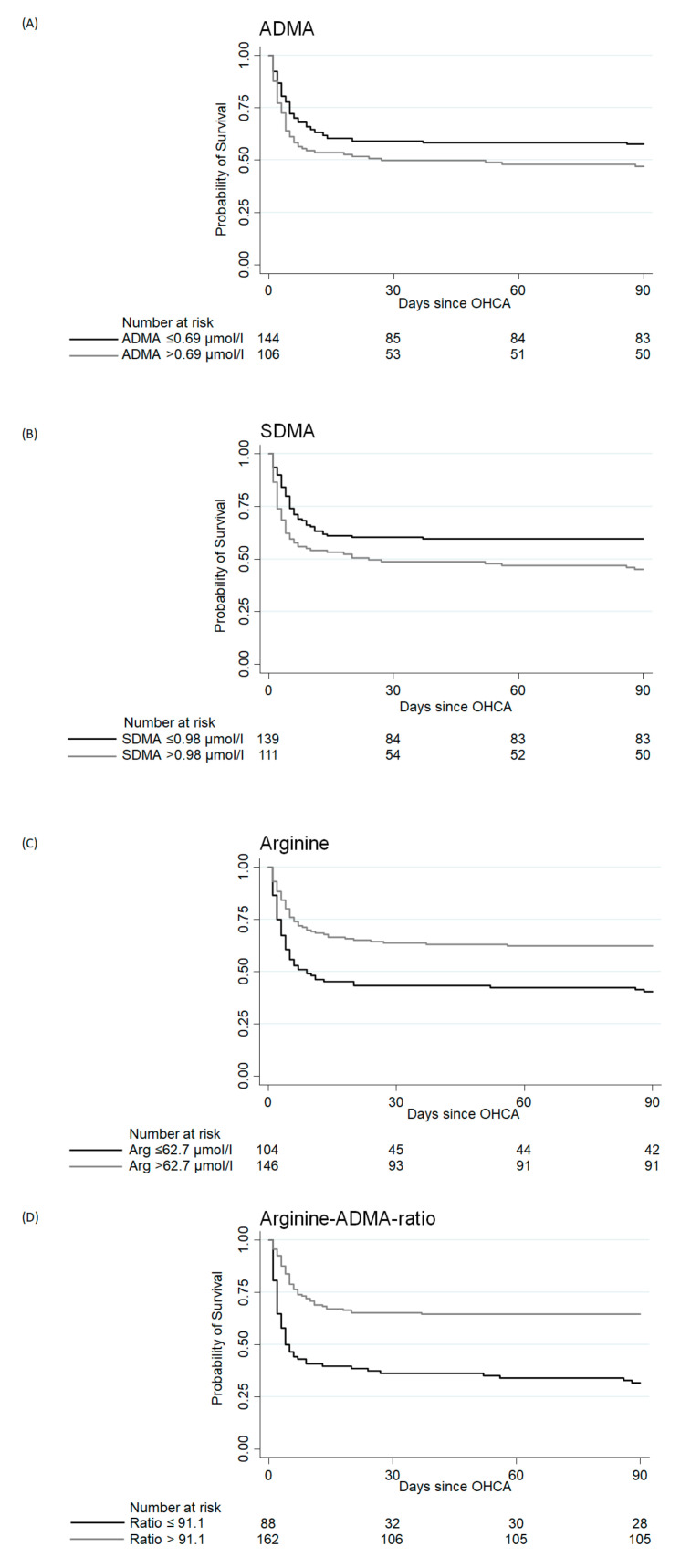
Kaplan-Meier survival estimates at 90-day mortality assessment looking at asymmetric dimethylarginine (ADMA), symmetric dimethylarginine (SDMA), Arginine and Arginine/ADMA ratio. Legend: (**A**) Kaplan-Meier survival estimates at 90-day mortality assessment looking at asymmetric dimethylarginine (ADMA). (**B**) Kaplan-Meier survival estimates at 90-day mortality assessment looking at symmetric dimethylarginine (SDMA). (**C**) Kaplan-Meier survival estimates at 90-day mortality assessment looking at Arginine. (**D**) Kaplan-Meier survival estimates at 90-day mortality assessment looking at Arginine/ADMA ratio; A-D: Days since OHCA are shown on the x-axis, the y-axis displays the probability of survival.

**Figure 2 jcm-09-03815-f002:**
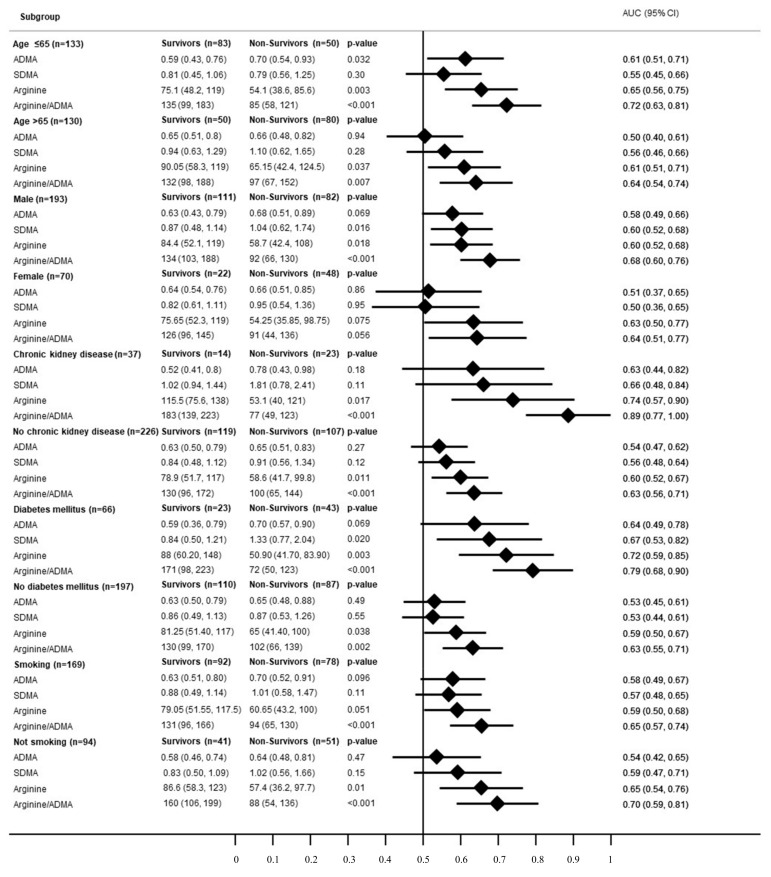
Forest plot for subgroups in regard of the primary endpoint (90-day mortality)**.** Legend: Forest plot for subgroups in regard of the primary endpoint (90-day mortality). Data is presented as area under the curve (AUC) and 95% confidence interval (95%CI). ADMA: asymmetric dimethylarginine; SDMA: symmetric dimethylarginine.

**Table 1 jcm-09-03815-t001:** Baseline characteristics of all patients and stratified by 90-day survival.

Factor	All	Survivors	Non-Survivors	*p*-Value
Number (*n*)	263	133	130	
Sociodemographics				
Age, median (IQR)	65 (57, 74)	62 (53, 73)	69 (61, 78)	<0.001
Male sex, *n* (%)	193 (73.4%)	111 (83.5%)	82 (63.1%)	<0.001
Cause of cardiac arrest				
Coronary artery disease, *n* (%)	130 (49.6%)	79 (59.8%)	51 (39.2%)	<0.001
Arrhythmia, *n* (%)	55 (21.0%)	28 (21.2%)	27 (20.8%)	0.93
Other / unknown, *n* (%)	77 (29.4%)	25 (18.9%)	52 (40.0%)	<0.001
Comorbidities				
Chronic kidney disease, *n* (%)	37 (14.1%)	14 (10.6%)	23 (17.7%)	0.10
Chronic obstructive pulmonary disease, *n* (%)	19 (7.3%)	7 (5.3%)	12 (9.2%)	0.22
Congestive heart failure, *n* (%)	38 (14.5%)	16 (12.1%)	22 (16.9%)	0.27
Coronary artery disease, *n* (%)	180 (68.7%)	101 (76.5%)	79 (60.8%)	<0.01
Cardiovascular risk factors				
Positive smoking history, *n* (%)	132 (61.1%)	78 (66.7%)	54 (54.5%)	0.07
Hypertension, *n* (%)	139 (53.1%)	70 (53.0%)	69 (53.1%)	0.99
Positive family history, *n* (%)	70 (35.2%)	40 (37.4%)	30 (32.6%)	0.48
Hyperlipidemia, *n* (%)	94 (42.5%)	51 (44.7%)	43 (40.2%)	0.49
Diabetes mellitus, *n* (%)	66 (25.2%)	23 (17.4%)	43 (33.1%)	<0.01
Resuscitation Information				
Time to ROSC (min), median (IQR)	20.5 (12, 32)	15 (10, 25)	29 (19, 40)	<0.001
Bystander CPR, *n* (%)	170 (64.9%)	99 (74.4%)	71 (55.0%)	<0.01
Initial shockable rhythm, *n* (%)	159 (63.3%)	104 (81.9%)	55 (44.4%)	<0.001
Use of epinephrine during CPR, *n* (%)	159 (63.3%)	61 (47.7%)	98 (79.7%)	<0.001
Initial ICU treatment and status				
Targeted temperature management (TTM), *n* (%)	165 (63.0%)	78 (59.1%)	87 (66.9%)	0.19
Intubation, *n* (%)	225 (85.9%)	99 (75.0%)	126 (96.9%)	<0.001
Vasoactive drugs, *n* (%)	186 (71.0%)	83 (62.9%)	103 (79.2%)	<0.01
Mean arterial pressure (mmHg), median (IQR)	82 (70, 93)	84 (71.5, 93)	80 (70, 93)	0.19
Systolic blood pressure (mmHg), median (IQR)	116 (101, 130)	117 (101, 128.5)	116 (101, 130)	0.82
Diastolic blood pressure (mmHg), median (IQR)	67 (55, 77)	70 (58, 79)	64.5 (51, 77)	0.04
Heartrate (bpm), median (IQR)	85 (74, 99)	81 (69.5, 94.5)	88.5 (76, 102)	0.02
Respiratory rate, median (IQR)	17 (14, 20)	17 (14, 20)	16 (14, 20)	0.90
Initial laboratory parameters				
pH, median (IQR)	7.27 (7.2, 7.3)	7.28 (7.2, 7.3)	7.24 (7.1, 7.3)	0.01
Lactate (mmol/l), median (IQR)	6.2 (3.6, 9)	4.8 (2.8, 6.9)	7.6 (5.2, 10)	<0.001
Creatinine (µmol/l), median (IQR)	99 (78, 122)	92.5 (78, 108)	109 (76, 144)	<0.01
PCT (ng/mL), median (IQR)	0.52 (0.15, 2.14)	0.19 (0.1, 0.84)	1.26 (0.4, 3.91)	<0.001
WBC (g/l), median (IQR)	14 (10.3, 18.9)	13 (9.64, 16.2)	15.4 (11, 20.3)	<0.01
CRP (mg/dl), median (IQR)	5.9 (1.5, 20.7)	3 (1.25, 10.8)	12.2 (2.6, 33.3)	<0.001

Legend: Data presented as *n* (%) or median (interquartile range, IQR). ROSC, return of spontaneous circulation; CPR, cardiopulmonary resuscitation; ICU, intensive care unit; PCT, procalcitonin; WBC, white blood cell count; CRP, C-reactive protein.

**Table 2 jcm-09-03815-t002:** Univariable regression analyses for 90-day mortality, in-hospital mortality and neurological outcome at hospital discharge.

Primary Endpoint: 90-Day Mortality						
	**Survivors (*n* = 133)**	**Non-Survivors (*n* = 130)**	***p*** **-Value**	**OR or Coefficient (95% CI)**	***p*** **-Value**	**AUC**
ADMA (µmol/L)	0.63 (0.50, 0.79)	0.67 (0.51, 0.88)	0.09	2.35 (1.08, 5.11)	0.03	0.56
ADMA log				1.75 (0.98, 3.11)	0.06	
ADMA per decile				1.07 (0.99, 1.17)	0.10	
SDMA (µmol/L)	0.85 (0.50, 1.13)	1.02 (0.58, 1.56)	0.03	1.06 (0.87, 1.29)	0.59	0.58
SDMA log				1.34 (0.99, 1.82)	0.06	
SDMA per decile				1.10 (1.01, 1.20)	0.03	
Arginine (µmol/L)	82 (52.3, 119)	58.1 (41.7, 99.8)	<0.01	0.57 (0.34, 0.96)	0.03	0.61
Arginine log				0.51 (0.34, 0.76)	<0.01	
Arginine per decile				0.87 (0.80, 0.95)	<0.01	
Ratio	134 (99, 183)	91 (64, 134)	<0.001	0.64 (0.47, 0.88)	<0.01	0.67
Ratio log				0.40 (0.26, 0.61)	<0.001	
Ratio per decile				0.80 (0.73, 0.88)	<0.001	
**Secondary endpoint: in-hospital mortality**						
	**Survivors (*n* = 145)**	**Non-Survivors (*n* = 118)**	***p*** **-Value**	**OR or Coefficient (95% CI)**	***p*** **-Value**	**AUC**
ADMA (µmol/L)	0.63 (0.50, 0.79)	0.67 (0.50, 0.88)	0.14	2.13 (1.01, 4.50)	0.047	0.55
ADMA log				1.64 (0.92, 2.91)	0.09	
ADMA per decile				1.06 (0.98, 1.16)	0.16	
SDMA (µmol/L)	0.89 (0.51, 1.15)	0.97 (0.55, 1.56)	0.18	1.03 (0.85, 1.25)	0.76	0.55
SDMA log				1.20 (0.89, 1.63)	0.23	
SDMA per decile				1.06 (0.98, 1.16)	0.15	
Arginine (µmol/L)	77.5 (51.4, 117)	58.2 (41.8, 110)	0.02	0.71 (0.43, 1.18)	0.19	0.59
Arginine log				0.60 (0.42, 0.87)	<0.01	
Arginine per decile				0.90 (0.83, 0.98)	0.02	
Ratio	130 (95, 181)	92 (65, 138)	<0.001	0.75 (0.55, 1.01)	0.06	0.63
Ratio log				0.50 (0.34, 0.74)	<0.001	
Ratio per decile				0.85 (0.78, 0.93)	<0.001	
**Secondary endpoint: Neurological outcome (CPC) at hospital discharge**						
	**Good Neurological Outcome (*n* = 122)**	**Poor Neurological Outcome (*n* = 141)**	***p*** **-Value**	**OR or Coefficient (95%CI)**	***p*** **-Value**	**AUC**
ADMA (µmol/L)	0.65 (0.51, 0.80)	0.65 (0.49, 0.83)	0.53	1.73 (0.82, 3.64)	0.15	0.52
ADMA log				1.33 (0.75, 2.33)	0.33	
ADMA per decile				1.03 (0.94, 1.12)	0.52	
SDMA (µmol/L)	0.91 (0.52, 1.18)	0.91 (0.54, 1.46)	0.54	0.99 (0.82, 1.21)	0.96	0.52
SDMA log				1.02 (0.76, 1.37)	0.89	
SDMA per decile				1.03 (0.95, 1.12)	0.45	
Arginine (µmol/L)	83.2 (52.5, 119)	58.6 (41.8, 108)	<0.01	0.57 (0.34, 0.96)	0.03	0.61
Arginine log				0.51 (0.34, 0.77)	<0.01	
Arginine per decile				0.87 (0.80, 0.95)	<0.01	
Ratio	131 (98, 184)	100 (64, 139)	<0.001	0.73 (0.54, 0.97)	0.03	0.65
Ratio log				0.47 (0.31, 0.70)	<0.001	
Ratio per decile				0.83 (0.76, 0.91)	<0.001	

Legend: Data presented as median (interquartile range). ADMA: asymmetric dimethylarginine; SDMA: symmetric dimethylarginine; Ratio: Arginine/ADMA ratio, log: log transformed with a base of 10, OR, odds ratio. OR of Arginine and Arginine/ADMA ratio relates to a 100 unit increase in blood level. CPC: Cerebral Performance Category Scale.

**Table 3 jcm-09-03815-t003:** Multivariable regression analyses for 90-day mortality, in-hospital mortality and neurological outcome at hospital discharge.

Primary Endpoint: 90-Day Mortality				
	(A) Adjusted for Age, Sex and Comorbidities		(B) Adjusted for Age, Sex and Resuscitation Circumstances	
	OR or Coefficient (95% CI)	*p-*Value	OR or Coefficient (95% CI)	*p-*Value
ADMA	2.44 (1.09, 5.45)	0.03	3.58 (1.25, 10.24)	0.017
ADMA log	1.72 (0.93, 3.21)	0.09	2.36 (1.10, 5.08)	0.028
ADMA per decile	1.07 (0.97, 1.17)	0.17	1.11 (0.99, 1.24)	0.07
SDMA	0.99 (0.80, 1.22)	0.90	1.01 (0.80, 1.28)	0.912
SDMA log	1.17 (0.84, 1.65)	0.36	1.47 (1.00, 2.15)	0.048
SDMA per decile	1.07 (0.97, 1.17)	0.18	1.14 (1.02, 1.28)	0.018
Arginine	0.54 (0.31, 0.94)	0.03	0.45 (0.24, 0.85)	0.014
Arginine log	0.47 (0.31, 0.73)	<0.01	0.45 (0.27, 0.76)	<0.01
Arginine per decile	0.85 (0.77, 0.94)	<0.01	0.88 (0.80, 0.98)	<0.01
Ratio	0.64 (0.46, 0.88)	<0.01	0.55 (0.39, 0.78)	0.001
Ratio log	0.40 (0.25, 0.62)	<0.001	0.32 (0.18, 0.54)	<0.001
Ratio per decile	0.80 (0.72, 0.88)	<0.001	0.77 (0.69, 0.87)	<0.001
**Secondary endpoint: In-hospital mortality**				
	**(A) Adjusted for age, sex and comorbidities**		**(B) Adjusted for age, sex and resuscitation circumstances**	
	**OR or Coefficient (95% CI)**	***p-*Value**	**OR or Coefficient (95% CI)**	***p-*Value**
ADMA	1.98 (0.90, 4.36)	0.09	3.00 (1.06, 8.52)	0.039
ADMA log	1.46 (0.78, 2.72)	0.23	2.17 (1.00, 4.69)	0.049
ADMA per decile	1.04 (0.94, 1.14)	0.45	1.09 (0.98, 1.22)	0.113
SDMA	1.00 (0.81, 1.24)	0.97	0.97 (0.78, 1.22)	0.81
SDMA log	1.10 (0.78, 1.54)	0.59	1.26 (0.87, 1.83)	0.224
SDMA per decile	1.04 (0.94, 1.14)	0.44	1.09 (0.97, 1.21)	0.136
Arginine	0.63 (0.36, 1.09)	0.10	0.62 (0.33, 1.17)	0.142
Arginine log	0.55 (0.36, 0.83)	<0.01	0.60 (0.37, 0.96)	0.034
Arginine per decile	0.88 (0.80, 0.96)	<0.01	0.90 (0.81, 1.00)	0.057
Ratio	0.72 (0.53, 0.98)	0.03	0.67 (0.48, 0.92)	0.013
Ratio log	0.50 (0.33, 0.75)	<0.001	0.46 (0.28, 0.76)	0.002
Ratio per decile	0.85 (0.77, 0.93)	<0.001	0.83 (0.75, 0.93)	0.001
**Secondary endpoint: Neurological Outcome (CPC) at hospital discharge**				
	**(A) Adjusted for age, sex and comorbidities**		**(B) Adjusted for age, sex and resuscitation circumstances**	
	**OR or Coefficient (95% CI)**	***p-*Value**	**OR or Coefficient (95% CI)**	***p-*Value**
ADMA	1.64 (0.76, 3.55)	0.21	2.42 (0.82, 7.16)	0.109
ADMA log	1.18 (0.64, 2.18)	0.59	1.61 (0.72, 3.57)	0.246
ADMA per decile	1.01 (0.92, 1.10)	0.91	1.04 (0.93, 1.17)	0.491
SDMA	0.91 (0.74, 1.13)	0.41	0.93 (0.74, 1.17)	0.524
SDMA log	0.84 (0.60, 1.16)	0.29	1.01 (0.68, 1.50)	0.953
SDMA per decile	0.99 (0.90, 1.08)	0.77	1.05 (0.93, 1.17)	0.451
Arginine	0.53 (0.31, 0.93)	0.03	0.43 (0.21, 0.86)	0.018
Arginine log	0.48 (0.31, 0.75)	<0.01	0.44 (0.25, 0.77)	0.004
Arginine per decile	0.85 (0.77, 0.94)	<0.01	0.85 (0.75, 0.96)	0.007
Ratio	0.72 (0.54, 0.97)	0.03	0.58 (0.41, 0.81)	0.001
Ratio log	0.47 (0.31, 0.72)	<0.01	0.36 (0.20, 0.63)	<0.001
Ratio per decile	0.83 (0.76, 0.92)	<0.001	0.81 (0.71, 0.91)	<0.001

Legend: Model (**A**) comorbidities: congestive heart failure, COPD, neurological disease, liver failure, chronic kidney failure, diabetes mellitus. Model (**B**) resuscitation circumstances: time until ROSC, witnessed cardiac arrest, bystander CPR, shockable rhythm, use of epinephrine during CPR. Ratio: Arginine/ADMA ratio OR of Arginine and Arginine/ADMA ratio relates to a 100 unit increase in blood level. ADMA: asymmetric dimethylarginine; SDMA: symmetric dimethylarginine; Ratio: Arginine/ADMA ratio; log: log transformed with a base of 10; OR: odds ratio; CPC: Cerebral Performance Category Scale.

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
