# Peer review of "Arginine and Arginine/ADMA Ratio Predict 90-Day Mortality in Patients with Out-of-Hospital Cardiac Arrest—Results from the Prospective, Observational COMMUNICATE Trial"

_jcm, 2020, doi:10.3390/jcm9123815_

Round 1

Reviewer 1 Report

In this manuscript, authors are reporting the role of arginine, SDMA, ADMA, and arginine/ADMA ratio in predicting the mortality in out-of-hospital cardiac arrest (OHCA) patients. There are some reports which assessed the role of ADMA, SDMA, and arginine/ADMA ratio in predicting the risk of cardiovascular events in patients with coronary artery disease (Schnabel et al. Circ. Res., 2005,97:e53-59; Bode-Boger et al. J Am Soc Nephrol. 2006,17:1128-1134; Yu et al. J Clin Endocrinol Metab. 2017,102:1879–1888). Either high levels of ADMA was associated with increased cardiovascular risk or higher arginine/ADMA ratio was associated with lower incidence of cardiovascular disease. In the current study, authors have measured the serum/plasma levels of arginine, SDMA, ADMA; and determined the arginine/ADMA ratio to predict mortality in OHCA patients. They have found that decreased arginine, elevated ADMA & SDMA, and low arginine/ADMA ratio were associated with mortality in OHCA patients.

 Comments:

  1. Page 2, line 60: Although there are several studies (e.g., Schnabel et al. Circ. Res., 2005,97:e53-59; Bode-Boger et al. J Am Soc Nephrol. 2006,17:1128-1134; Yu et al. J Clin Endocrinol Metab. 2017,102:1879–1888) assessing the role of arginine, ADMA, SDMA, and arginine/ADMA in cardiovascular diseases, authors have cited only one study [ref 15]. I would suggest authors to include more citations on the role of arginine, ADMA, SDMA, and arginine/ADMA in cardiovascular diseases.
  2. Page 2, line 88: Provide more details on blood samples collection. It is not clear if the authors used plasma or serum for analysis. The abstract says serum while the methods section says plasma. Please clarify. At what temperature, the samples were stored? -20 or -80 C?
  3. Page 2, line 90: No details have been provided on the quantification of arginine, ADMA or SDMA using LC-MS. Briefly describe the LC-MS methods used for the quantification of these metabolites, quoting relevant references.
  4. Page 4, line 140 (Table 2): It will be easy to visualize the quantitative data in the form of a plot/diagram. Authors may provide the levels of arginine, ADMA, SDMA and arginine/ADMA (Survivors vs. Non-survivors) in the form of a box-and-whisker plot.
  5. Page 4, Discussion section: I suggest the authors to start the discussion with the role of arginine, ADMA, SDMA and arginine/ADMA in cardiovascular diseases citing relevant references.

Author Response

Dear Prof. Andrès, dear Prof. Hennerici, dear Reviewers,

We would like to thank again for your valuable time and input.  We have carefully considered your comments and addressed them in a revised version of our manuscript.  Below this letter, please find a point-by-point description of how these comments were addressed. We hope that you will now find to merit publication in the Journal of Clinical Medicine

The authors have declared that no competing interests exist.

Sincerely,

Christoph Becker and Sabina Hunziker on behalf of all authors

Reviewer 1

Page 2, line 60: Although there are several studies (e.g., Schnabel et al. Circ. Res., 2005,97:e53-59; Bode-Boger et al. J Am Soc Nephrol. 2006,17:1128-1134; Yu et al. J Clin Endocrinol Metab. 2017,102:1879–1888) assessing the role of arginine, ADMA, SDMA, and arginine/ADMA in cardiovascular diseases, authors have cited only one study [ref 15]. I would suggest authors to include more citations on the role of arginine, ADMA, SDMA, and arginine/ADMA in cardiovascular diseases.

Authors: Thank you for this valuable input. We have added these citations as suggested.

Page 2, line 88: Provide more details on blood samples collection. It is not clear if the authors used plasma or serum for analysis. The abstract says serum while the methods section says plasma. Please clarify. At what temperature, the samples were stored? -20 or -80 C?

Authors: Thank you for your enquiry. We have used plasma for analysis and corrected the abstract correspondingly.

We stored the plasma samples at a temperature of -80°C. We have added a statement for clarification at line 91.  

Page 2, line 90: No details have been provided on the quantification of arginine, ADMA or SDMA using LC-MS. Briefly describe the LC-MS methods used for the quantification of these metabolites, quoting relevant references.

Authors: We have elaborated this method a little bit more in detail and have added a  relevant reference stating the reliability of this method in measuring metabolomics stating “They were later analyzed quantitatively by liquid chromatography coupled to tandem mass spectrometry (LC-MS/MS) using an Ultimate 3000 UHPLC (Thermo Fisher, San Jose, USA) system coupled to an ABSciex QTRAP 5500 quadrupole mass spectrometer (ABSciex, Darmstadt, Germany) and the AbsoluteIDQ p180 Kit (BIOCRATES Life Sciences AG, Innsbruck, Austria)[34-36]. The Biocrates AbsoluteIDQ™ p180 kit is a commercially available targeted metabolomics assay that can be used on a variety of LC-MS/MS triple quadrupole instruments. A recent interlaboratory assessment of this metabolomic assay showed that this method delivers reliable and reproducible results[37].”

Page 4, line 140 (Table 2): It will be easy to visualize the quantitative data in the form of a plot/diagram. Authors may provide the levels of arginine, ADMA, SDMA and arginine/ADMA (Survivors vs. Non-survivors) in the form of a box-and-whisker plot.

Authors: Thank you for this suggestion. We have created a box plot to visualize the plasma levels of ADMA, SDMA, Arginine and Arginine/ADMA ratio in survivors and non-survivors. If you agree, we would to like to add this plot to the Appendix of our paper.

Page 4, Discussion section: I suggest the authors to start the discussion with the role of arginine, ADMA, SDMA and arginine/ADMA in cardiovascular diseases citing relevant references.

Authors: Thank you for your input. We have changed the discussion as suggested and start now with discussing the role of arginine and its metabolites in patients with cardiovascular diseases: “ First, patients with diseases caused by endothelial dysfunction as well as cardiovascular risk factors appear to have elevated circulating levels of ADMA[10,11,18].{Cook, 2000 #48}{Cook, 2000 #48} Furthermore, the risk of cardiovascular events and death is increased in patient with elevated levels of ADMA and SDMA[15,17,18]. There is evidence that in patients with ischemic heart disease and cardiogenic shock, which is one of the most common underlying etiologies of cardiac arrests, high levels of ADMA and low Arginine/ADMA ratios are associated with impaired myocardial perfusion, hemodynamic instabilities as well as with short and long-term mortality[24,40]. These pathophysiologic mechanisms might explain why Arginine/ADMA ratio and Arginine are predictive for 90d mortality in our cohort.”

Reviewer 2 Report

Thank you so much for giving me an opportunity to peer review the manuscript entitled “Arginine and Arginine/ADMA Ratio Predict 90-day 2 Mortality in Patients with Out-of-hospital Cardiac Arrest: Results from the prospective, observational COMMUNICATE trial”. I’ve read it with great interest. I think the manuscript is well-written, but I have several critical concerns.

Major

  1. The authors report that the arginine/ADMA ratio (and other biomarkers) predicts 90-day mortality (and other outcomes), with an AUC of 0.67 for 90-day mortality. One concern is that the arginine/ADMA ratio is not a routinely measured biomarker, and besides, its predictive potential is not so high. Is arginine/ADMA ratio better than other simple laboratory tests? For example, lactate is easy-to-be-measured biomarker and reported to be associated with clinical outcomes in OHCA patients. Additionally, lactate is metabolized by the liver and is largely unaffected by chronic kidney disease. (Moreover, hyperlactatemia is usually mild with chronic liver disease). Thus, I need the authors to compare the arginine/ADMA ratio to lactate or other simple biomarkers. It would be OK that they perform some additional analyses or discuss in the discussion section.

  1. According to Table 1, almost 40% of patients didn’t receive targeted temperature management. Could the authors explain the reason? Moreover, I’d like them to mention how to perform their TTM (target temperature, maintain time etc)

  1. According to Table 3, the authors have performed 12 multivariable analyses per each model. Shouldn't multiple correction be done? I wonder if there are some alpha errors. Wouldn't it be better to consider one solid model rather than many models?

  1. I am concerned that important pre-hospital variables including initial rhythm, witness status, bystander CPR etc are not treated as covariates in multivariate analysis. How did the authors select covariates?

  1. The authors have discussed the sex difference in the discussion area. However, they did not show P for interaction.

Minor

  1. I’ve got an impression that the inclusion of 263 people between November 2012 and June 2018 seemed a bit too small. If there were patients excluded, I need the authors to clearly state the inclusion and exclusion criteria.

Author Response

Dear Prof. Andrès, dear Prof. Hennerici, dear Reviewers,

We would like to thank again for your valuable time and input.  We have carefully considered your comments and addressed them in a revised version of our manuscript.  Below this letter, please find a point-by-point description of how these comments were addressed. We hope that you will now find to merit publication in the Journal of Clinical Medicine

The authors have declared that no competing interests exist.

Sincerely,

Christoph Becker and Sabina Hunziker on behalf of all authors

Reviewer 2

The authors report that the arginine/ADMA ratio (and other biomarkers) predicts 90-day mortality (and other outcomes), with an AUC of 0.67 for 90-day mortality. One concern is that the arginine/ADMA ratio is not a routinely measured biomarker, and besides, its predictive potential is not so high. Is arginine/ADMA ratio better than other simple laboratory tests? For example, lactate is easy-to-be-measured biomarker and reported to be associated with clinical outcomes in OHCA patients. Additionally, lactate is metabolized by the liver and is largely unaffected by chronic kidney disease. (Moreover, hyperlactatemia is usually mild with chronic liver disease). Thus, I need the authors to compare the arginine/ADMA ratio to lactate or other simple biomarkers. It would be OK that they perform some additional analyses or discuss in the discussion section.

Authors: Thank you very much for this input. We agree that Arginine and its metabolites are markers not routinely measured or whose measurement is not always available. We also agree that an AUC of 0.67 is moderate. However, the predictive potential of lactate regarding our primary endpoint of 90-day mortality was similar for lactate with an AUC of 0.69. We have added a corresponding statement to the end of our limitation section: “Last, arginine/ADMA ratio is not a routinely measured biomarker and with an AUC of 0.67 its use might be limited in clinical routine. However, the predictive potential of lactate, a routine blood marker reported to be associated with clinical outcomes in OHCA patients, was similar with an AUC of 0.69.”

According to Table 1, almost 40% of patients didn’t receive targeted temperature management. Could the authors explain the reason? Moreover, I’d like them to mention how to perform their TTM (target temperature, maintain time etc)

Authors: Thank you for your enquiry. The reason why a substantial proportion of patients did not receive targeted temperature management in our study was that initially TTM was only given to patients with a shockable rhythm. Today, it is offered to all patients after cardiac arrest. Above, if patients were conscious after cardiac arrest, TTM was not initiated.

We now added the following text to better explain TTM used during the trial  “The treatment of patients regarding the cardiac arrest was based on the clinical routine in our intensive care unit without interaction with the research team. In 2012, all consecutive patients without complete recovery to premorbid neurofunctional baseline within the first hour following resuscitation were treated with in-hospital systemic cooling via the thermogard XP temperature management system (ZOLL® Medical Corporation, Chelmsford, MA, USA) as a neuroprotectant measure to a target core temperature of 93.2 degrees Fahrenheit (i.e., 34.0 degrees Celsius) for 24 hours followed by a rewarming phase with a controlled increase of the core temperature (i.e., 0.2 degrees Fahrenheit or 0.1 degrees Celsius) per hour until 99.5 degrees Fahrenheit (i.e., 37.5 degrees Celsius). Since 2013 (following the TTM-trial 35), all consecutive patients without complete recovery were cooled to a target core temperature of 96.8 degrees Fahrenheit (i.e., 36.0 degrees Celsius) for 28 hours followed by the rewarming phase using the same thermogard XP temperature management system as mentioned above. Patient with core temperatures below the target temperature were rewarmed with 32.9 degrees Fahrenheit (i.e., 0.5 degrees Celsius) to meet the target core temperatures.”.

I am concerned that important pre-hospital variables including initial rhythm, witness status, bystander CPR etc are not treated as covariates in multivariate analysis. How did the authors select covariates?

Authors: Thanks for your valuable input.  We did not include all potential variables to prevent collinearity, however we have now adapted the multivariate model as suggested and now adjust for resuscitation circumstances including time until ROSC, witnessed cardiac arrest, bystander CPR, shockable rhythm, use of epinephrine during CPR.  

According to Table 3, the authors have performed 12 multivariable analyses per each model. Shouldn't multiple correction be done? I wonder if there are some alpha errors. Wouldn't it be better to consider one solid model rather than many models?

Authors: Thank you for this feedback. We had predefined models to look at different aspects of the cardiac arrest and resuscitation circumstances. However, as mentioned above, we have incorporated your input into a solid model (Adjusted for age, sex and resuscitation circumstances) and, if you agree, would like to shift the other models (“adjusted for age, gender and cardiovascular risk factors” and “adjusted for age, gender and laboratory values”) into the appendix.

The authors have discussed the sex difference in the discussion area. However, they did not show p for interaction.

Authors: Thank you for pointing this out. We agree with you, and now included a p for interaction and revised the paragraph in the discussion as following: “Fifth, we assessed potential sex differences regarding the predictive value of Arginine and its metabolites in our study. While Arginine and Arginine/ADMA ratio were significantly associated with 90-day mortality in male patients, these associations were not found in female patients. However, there was no strong evidence for effect modification by gender (p of interaction =0.10).»

Minor

I’ve got an impression that the inclusion of 263 people between November 2012 and June 2018 seemed a bit too small. If there were patients excluded, I need the authors to clearly state the inclusion and exclusion criteria.

Authors: Thank you for your input. We have now stated the inclusion criteria more clearly. Additionally, we also had some patients or their families refusing to participate in the study. We also added now a statement in the limitation section as follows “Also, some family members refused to give informed consent to participate in the study and there is thus potential for selection bias.”.

Round 2

Reviewer 1 Report

The authors have done a nice job in revising the manuscript. I have one minor suggestion regarding the quantification of Arginine, ADMA, SDMA using the Biocrates AbsoluteIDQ® p180 Kit. It is better to add a sentence (page 3, line 114) saying that "arginine, ADMA, and SDMA are part of the AbsoluteIDQ® p180 Kit quantified by LC-MS".

I am fine with authors in providing the box-plot of arginine, ADMA, Arginine/ADMA, and SDMA of survivors and non-survivors in the Appendix.

Reviewer 2 Report

There are still a few more questions I would like to ask the authors personally, but as a reviewer I have nothing more to add. I felt their sincerity and effort in the response letter and revisions.